# A Low-Cost, High-Fat Diet Effectively Induces Obesity and Metabolic Alterations and Diet Normalization Modulates Microbiota in C57BL/6 Mice

**DOI:** 10.3390/nu17233806

**Published:** 2025-12-04

**Authors:** Iasmim Xisto Campos, Marcella Duarte Villas Mishima, Fermín I. Milagro, Maria do Carmo Gouveia Peluzio

**Affiliations:** 1Department of Nutrition and Health, Federal University of Viçosa—UFV, University Campus, Viçosa 36570-900, MG, Brazil; iasmim.campos@ufv.br (I.X.C.); marcella.mishima@ufv.br (M.D.V.M.); 2Center for Nutrition Research, Department of Nutrition, Food Science and Physiology, Faculty of Pharmacy and Nutrition, University of Navarra, 31008 Pamplona, Spain; fmilagro@unav.es; 3Navarra Institute for Health Research (IdiSNA), 31008 Pamplona, Spain; 4Centro de Investigación Biomédica en Red de la Fisiopatología de la Obesidad y Nutrición (CIBEROBN), Instituto de Salud Carlos III, 28029 Madrid, Spain

**Keywords:** experimental model, oxidative stress, butyrate, dysbiosis

## Abstract

**Background/Objectives**: High-fat diets (HFDs) are widely used to induce obesity, but cost-effective and reproducible formulations remain challenging. Moreover, the reversibility of metabolic and gut microbiota alterations following HFD withdrawal is not fully understood. This study evaluated a low-cost HFD model in mice and investigated metabolic, oxidative, and gut microbiota changes during a subsequent 12-week dietary normalization phase. **Methods**: Male C57BL/6 mice were fed a standard diet (CTN) or a lard-supplemented HFD for 12 weeks (Phase 1), followed by 12 weeks dietary normalization to a standard diet (Phase 2). Body weight, adiposity, blood glucose, biochemical parameters, and oxidative markers were assessed. Fecal samples were analyzed for short-chain fatty acids (SCFAs), microbiota composition (16S rRNA sequencing), and predicted functions using FAPROTAX and PICRUSt2. **Results**: The HFD significantly increased body weight, abdominal circumference, the Lee index, and adipose tissue mass compared to CTN. Following diet normalization, both groups exhibited weight loss, but the previously obese mice maintained a higher Lee index and distinct lipid and uric acid profiles. No hepatic oxidative stress was detected after normalization. SCFA profiles underwent a temporal shift: CTN showed higher fecal acetate, while HFD mice exhibited elevated butyrate. Functional prediction revealed one pathway associated with an unclassified Rickettsiales bacterium that was exclusively found in HFD mice. The CTN group exhibited a higher abundance of the thiamine diphosphate formation pathway (PWY-7357), suggesting enhanced oxidative metabolism. **Conclusions**: This low-cost HFD successfully induced obesity and dysbiosis. Dietary normalization resulted in a partial modulation of metabolic and microbial balance, thereby highlighting host–microbe metabolic plasticity.

## 1. Introduction

Obesity, considered a global public health problem and the 21st-century pandemic, is defined as the excessive or abnormal accumulation of body fat, resulting in a disproportionate increase in weight relative to height [1]. It is associated with an increased risk of several diseases, accounting for approximately 2.8 million deaths annually from noncommunicable diseases such as diabetes, cancer, digestive disorders, cardiovascular, neurological, and respiratory diseases [2,3]. Its development is strongly related to the imbalance between energy consumption and expenditure, as well as to changes in the composition and quality of the diet, as the high consumption of sugar and saturated fat of Western diet and genetics and environmental factors end up making the study of prevention and treatment measures even more difficult [4,5]. Consequently, experimental models that reproduce metabolic and physiological aspects of human obesity are used for years [6,7,8].

The diet-induced obesity (DIO) model in C57BL/6 mice is one of the most widely used in metabolic research because these animals are highly susceptible to the metabolic changes observed in human obesity, including weight gain, insulin resistance, systemic inflammation, and hepatic alterations [9]. Traditionally, these protocols employ commercial purified diets providing 45–60% of calories from fat, often derived from animal sources (e.g., lard, beef tallow, coconut oil, or milk fat), to mimic the excessive lipid intake typical of the modern human diet [8,10]. The development of these metabolic alterations typically requires consuming an obesogenic diet for 12 [11] to 14 weeks [9]. Other protocols also utilize diets such as the Western diet, high-fructose, high-fat/high-fructose (HFHF) diet, or high-fat/high-sucrose (HFHS) diet [12].

However, the elaboration or acquisition of these commercial formulations can be a significant limitation for some laboratories, particularly in developing countries, restricting both reproducibility and access to experimental obesity models. Consequently, the development of low-cost HFDs, prepared by supplementing commercial chow with accessible lipid sources (such as lard), emerges as a viable and economically sustainable alternative for inducing experimental obesity [8].

Therefore, the present study aimed to evaluate the effect of a low-cost, easily reproducible high-fat diet (providing 45% of total calories from fat) on obesity induction in C57BL/6 mice. Furthermore, we analyzed the effects of a subsequent diet normalization phase on weight gain, adipose tissue accumulation, metabolic changes, and gut microbiota composition.

## 2. Materials and Methods

### 2.1. Preparation of Experimental Diets

To prepare the high-fat diet, the commercial diet (Neovia Nutrição e Saúde Animal LTDA, Descalvado, Brazil) based on AIN93M diet, a standard diet for rodents [13] with 3.14 kcal/g (Table 1), was ground in an industrial mill until powder and then lard was added (213 g/kg of diet) until elevating the diet to 45% of lipids and 4.17 kcal/g. After the incorporation, the pellets were produced, packed and frozen at −20 °C until offered to the animals.

### 2.2. Ethical Aspects

This work was approved by the Animal Use Ethics Committee of the Federal University of Viçosa (CEUA/UFV), under registration number 37/2022 and followed all the standards of the National Council for the Control of Animal Experimentation (CONCEA).

### 2.3. Experimental Design

The sample size was calculated as proposed by Mera et al., 1998 [14], adopting as a criterion a difference of 10% in weight gain (main variable) in relation to the treatments and a statistical power of 99% (α < 0.01). For the calculations, the values from the study by Miyoshi et al., 2014 [15] were used, totaling 8 animals per experimental group.

Sixteen male C57BL/6J mice, 30 days old, from the Central Animal Facility of the Center for Biological and Health Sciences, Federal University of Viçosa, were used. Initially, the animals remained for 4 weeks in an adaptation period, and subsequently, the experiment lasted 24 weeks [16]. The animals remained in the Experimental Nutrition Laboratory, Department of Nutrition and Health, in collective cages (4 animals/box), in an environment with controlled temperature (22 °C ± 2 °C) and a 12-h photo period. Distilled water and diet consumption was ad libitum throughout the experimental period.

The first phase of the experiment consisted of inducing obesity (Figure 1). After the adaptation period, the animals were weighed and randomly divided into two groups with the same average weight: a high-fat diet group (HFD) and a control group (CTN). The control group received the commercial diet, while the HFD group received the same commercial diet added with lard. This experimental phase lasted 12 weeks, with the diet offered ad libitum. At the end of this period, the animals were weighed and underwent biometric measures to identify the development of excess weight.

A second phase of the experiment was included to evaluate the effect of diet normalization based on the reduction in the amount of fat consumed by the animals. The animals in the HFD group were then fed the commercial diet, like the CTN group, for an additional 12 weeks. Diet consumption and weight assessments were performed weekly throughout the experimental period. Feces excreted by the animals on week 6 and 12 of Phase 2 were collected to assess short-chain fatty acids (SCFA) and metatoxonomy.

At the end of the 28-week period, the animals fasted for 8 h and were anesthetized for 2 to 4 min with 3% isoflurane (Isoflorine^®^, Cristália, Itapira, Brazil), using a simple circuit with a flowmeter attached to an oxygen cylinder. Blood samples were collected through an incision in the abdominal aorta, using Eppendorf tubes, and centrifuged at 12,000 rpm for 15 min. Liver, intestines and adipose tissue were washed in physiological saline solution, weighed and portioned, obtaining samples frozen in liquid nitrogen and stored at −80 °C for analysis.

### 2.4. Assessment of Weight and Body Measurements

The animals’ body weight was checked weekly using a digital scale, as was their food consumption. Body measurements, abdominal circumference and naso-anal length, were taken at the beginning and at the end of Phase 2 of the experiment (diet normalization) using an inelastic tape measure. The Lee index was calculated by dividing the cubic root of body weight (g) by the snout-anus length (cm) [17]. The relative weight was calculated by dividing the organ weight by body weight and multiplying the ratio by 100 [18].

### 2.5. Biochemical Analysis

Uric acid, creatinine, triglycerides, fasting blood glucose, total cholesterol and high-density lipoprotein (HDL-cholesterol) were determined in serum samples using colorimetric enzymatic methods using commercial kits (Bioclin^®^, Belo Horizonte, Brazil) according to the manufacturer’s procedures. The samples were analyzed on an automated analyzer (Mindray Medical International Limited, model BS 200, Shenzhen, China). Low-density lipoprotein (LDL-cholesterol) and very-low-density lipoprotein (VLDL) concentrations were calculated according to the equations proposed by Friedewald et al., 1972 [19].

### 2.6. Oxidative Stress

#### 2.6.1. Homogenate Preparation

To obtain a liver homogenate, 100 mg of liver sample was mixed with 1000 μL of phosphate buffer (50 mM). The sample was macerated and centrifuged at 12,000 rpm and 4 °C for 10 min. The supernatant was removed and stored in an ultrafreezer until the time of analysis.

#### 2.6.2. Superoxide Dismutase (SOD) Quantification

SOD activity was quantified based on the enzyme’s ability to inhibit pyrogallol auto-oxidation, as described by Dieterich et al., 2000 [20]. Measurements were performed in triplicate using a spectrophotometer (Multiskan GO, Thermo Scientific^®^, Ratastie, Finland) at 570 nm. Results are expressed as U of SOD per mg of protein, with protein concentration determined according to the method of Lowry et al., 1951 [21].

#### 2.6.3. Catalase (CAT) Activity

CAT activity was determined based on its ability to decompose hydrogen peroxide (H_2_O_2_) into water and molecular oxygen. Absorbance was measured at 0, 30, and 60 s at 240 nm. All readings were performed using a spectrophotometer (Multiskan GO, Thermo Scientific^®^, Ratastie, Finland) according to the method described by Aebi, 1984 [22]. Results are expressed as U of CAT per mg of protein, with protein concentration determined following the method of Lowry et al., 1951 [21].

#### 2.6.4. Lipid Peroxidation Measurement

The measurement of thiobarbituric acid-reactive substances (TBARS) was performed according to a method adapted from Buege and Aust, 1978 [23]. This assay is based on the ability of malondialdehyde (MDA), under acidic conditions and upon heating with TBARS, to form a pink-colored product that can be quantified using a spectrophotometer (Multiskan GO, Thermo Scientific^®^, Ratastie, Finland) at 535 nm. Liver homogenates were mixed with the TBARS solution, which contains trichloroacetic acid and thiobarbituric acid, and then incubated in a water bath at 100 °C for 10 min. The molar extinction coefficient of 1.56 × 10^5^ M^−1^ cm^−1^ was used to calculate TBARS concentration, and protein content was determined following the method of Lowry et al., 1951 [21]. Results are expressed in µmol/mg of protein.

#### 2.6.5. Protein Carbonyl

Protein oxidation was analyzed by quantifying carbonyls present in the pellets of homogenized tissues, using the 2,4-dinitrophenylhydrazine (DNPH) technique described by Levine et al., 1990 [24]. The measurements were performed using a spectrophotometer (ThermoScientific^®^, Multiskan GO model, Finland) at 370 nm. The results were expressed in nmol/mg protein.

### 2.7. Short Chain Fatty Acids (SCFA)

The extraction of SCFA (acetic, propionic, and butyric) from fecal content of excreted feces in the middle and the end of Phase 2 was performed according to the methodology proposed by Siegfried et al., 1984 [25]. They were analyzed in a high-performance liquid chromatograph (Dionex Ultimate 3000 Dual detector HPLC apparatus, Dionex Corporation ^®^, Sunnyvale, CA, USA). For chromatographic separation, the samples were injected into a column (RezexROA-Organic Acid H+ (8%), Phenomenex^®^, Torrance, CA, USA) with a length of 300 mm and a diameter of 7.8 mm. The mobile phase used was 5-molar sulfuric acid, injection flow of 0.7 mL/min, injection volume of 20 µL, and oven temperature of 45 °C. The detector used was the Rid RI-101 (Shodex^®^, Tokyo, Japan).

### 2.8. Composition of the Intestinal Microbiota

DNA was extracted from stool samples (100 ± 2 mg) using the E.Z.N.A.^®^ Stool DNA Kit (Omega Bio-tek Inc., Norcross, GA, USA), following the manufacturer’s instructions. DNA concentration, purity (260/230 and 260/280 ratios), and integrity were evaluated using a NanoDrop 1000 spectrophotometer (Thermo Scientific, Waltham, MA, USA) and successful amplification was confirmed by PCR. The V3–V4 hypervariable regions of the 16S rRNA gene were amplified using region-specific primers with Illumina (Illumina, San Diego, CA, USA) adapter overhangs, followed by indexing. Amplicons were size-selected using 2% agarose gel electrophoresis, pooled in equimolar amounts, end-repaired, A-tailed, ligated to Illumina adapters, and purified. Library quality was assessed by qPCR, and sequencing was performed by Novogene Co., Ltd. (Sacramento, CA, USA) on an Illumina paired-end platform (2 × 250 bp). The mean sequencing depth per sample ranged from 120,654 to 181,062 high-quality paired reads.

The raw data were filtered to remove low-quality and long reads, noise, and error correction. Then, the forward (F) and reverse (R) sequences were merged, followed by chimera removal in R Studio using the DADA2 package (v1.28.0) to infer amplicon sequence variants (ASVs). Chimeric sequences were identified and removed. Representative ASV sequences were taxonomically classified (phylum, family, and genus) using the Silva 16S rRNA database version 138-2.

Eukaryotic, mitochondrial, and chloroplast sequences were filtered, followed by the removal of sequences with fewer than 20 reads and a prevalence < 5%. Diversity indices were calculated using genus-level ASVs. Alpha diversity was calculated from a richness matrix using the Shannon, Simpson, and Chao1 indices. Beta diversity was analyzed by principal coordinates analysis (PCoA) using the Bray–Curtis dissimilarity index. The data were corrected using the FDR (false discovery rate) criterion.

To extrapolate the taxonomic composition of the microbial community into putative ecological and metabolic functions, we applied the Functional Annotation of Prokaryotic Taxa (FAPROTAX) pipeline [26]. This database maps taxonomic identities (mainly at the genus and species levels) to metabolic or ecological functions based on cultured representatives described in the literature. The analysis was performed using the FAPROTAX script (version 1.1) implemented in the cluster environment, according to the developer’s recommendations. We also analyzed functional inference from 16S rRNA data was performed using PICRUSt2 v2.5.0. ASV representative sequences were placed into a reference phylogeny with EPA-NG, and gene family abundances were predicted using hidden-state reconstruction. Predicted EC numbers, KEGG Orthologs and MetaCyc pathways were exported as unstratified metagenomes [27]. Functional prediction tables (pathways, KOs, and ECs) generated by PICRUSt2 were processed in R. Features were first filtered by prevalence (≥10% of samples) and mean abundance (≥20 counts) to remove extremely rare predictions that inflate the number of statistical tests without biological relevance. Data were normalized to relative abundance and matched to sample metadata. Outliers were removed prior to statistical testing using the IQR rule applied within each function, group, and timepoint. All statistical comparisons (intergroup and intragroup) were performed using Wilcoxon tests, and *p*-values were adjusted for multiple testing using the Benjamini–Hochberg false discovery rate (FDR) correction.

### 2.9. Statistical Analysis

Results are presented as mean ± standard deviation. The Shapiro–Wilk test was used to assess data normality. Comparisons between two independent groups were performed using Student’s *t*-test for normally distributed variables or the Mann–Whitney *U* test for non-normally distributed variables. When appropriate, paired comparisons between different time points were conducted using paired *t*-tests or Wilcoxon signed-rank tests. All statistical analyses were performed using GraphPad Prism version 8.0 (GraphPad Software, San Diego, CA, USA). Statistical significance was set at *p* < 0.05.

## 3. Results

### 3.1. Body and Metabolic Measurements

Diet consumption was higher in the HFD compared to CTN during the first five weeks of Phase 1. Consumption subsequently decreased in the HFD group (Figure 2A). Consequently, the HFD group’s caloric intake remained higher than the CTN group’s throughout Phase 1 (Figure 2B). The HFD group exhibited significantly greater body weight than the CTN group from the 4th week onwards (HFD: 25.88 g ± 1.82; CTN: 24.72 g ± 1.89), a divergence that continued until the end of Phase 1 (Figure 2C).

At the end of the induction period (Phase 1), the HFD group demonstrated a higher final body weight and gained more than three times the weight of the CTN group (Table 2). Consumption of the HFD for 12 weeks led to an increase not only in final body weight but also in final abdominal circumference, Lee index, and blood glucose levels in C57BL/6 mice (Table 2).

During Phase 2, after an additional 12 weeks of standard diet consumption, both groups showed weight loss, with the previously obese (HFD) group losing significantly more weight and experiencing a greater reduction in abdominal circumference (Δ) than the CTN group. Despite this weight loss, the HFD group maintained a significantly higher Lee index at the end of Phase 2 (Table 2).

At the end of Phase 2, the absolute and relative weights of the liver, small intestine, and colon did not differ between the groups (Figure 3). In contrast, despite having similar abdominal circumferences, the HFD group maintained significantly greater total adipose tissue mass at the end of Phase 2. Specifically, statistical differences were observed in the weights of the epididymal white adipose tissue (eWAT) (0.29 ± 0.07; 0.47 ± 0.11; *p* = 0.0040), retroperitoneal white adipose tissue (rpWAT) (0.13 ± 0.05; 0.31 ± 0.1; *p* = 0.0005), and total white adipose tissue (tWAT) (0.82 ± 0.14; 1.33 ± 0.26; *p* = 0.004) when comparing CTN and HFD, respectively.

Analysis of circulating metabolic variables in the blood at the end of Phase 2 revealed that the HFD group had significantly lower levels of uric acid (1.60 ± 0.19; 1.22 ± 0.11; *p* = 0.0370), total cholesterol (43.83 ± 2.56; 38.80 ± 3.90; *p* = 0.0298), and non-HDL cholesterol (17.57 ± 2.94; 8.44 ± 2.60; *p* = 0.0002) compared to the CTN group (Figure 4). Creatinine, triglycerides, fasting glucose and HDL cholesterol levels did not differ between groups.

### 3.2. Oxidative Stress in Liver

Regarding hepatic oxidative stress biomarkers, no statistically significant difference was observed between the CTN and HFD animals at the end of Phase 2 in the levels of the antioxidant enzymes Catalase (CAT) and Superoxide Dismutase (SOD), the lipid peroxidation marker TBARS, or protein carbonylation (Figure 5).

### 3.3. SCFA

Fecal SCFA content was evaluated at week 6 and week 12 of Phase 2. At week 6, the HFD group showed a lower amount of acetic acid compared to the CTN group (Figure 6). No differences were detected in butyric acid or propionic acid at this time point. At week 12, however, the HFD group exhibited a higher amount of butyric acid compared to the CTN group, with no differences found in acetic acid or propionic acid. Comparing the CTN group over time (intra-group comparison), a decrease in acetic acid was observed from Week 6 to Week 12, indicating a time-dependent effect of the standard diet.

### 3.4. Microbiota Analysis

Initially, we identified 9804 ASVs. After filtering, we obtained 891 ASVs that met the proposed requirements. Alpha diversity, assessed by the Simpson and Shannon indices, showed no significant differences in dominance or equitability between the groups at either Week 6 or Week 12 of Phase 2 (Figure 7A,B). Similarly, the Chao1 index for richness did not differ between groups (Figure 7C).

Permutational Multivariate Analysis of Variance (PERMANOVA) confirmed that there was no significant difference in the average composition of bacterial communities between the groups at either week 6 (*p* = 0.718) or week 12 (*p* = 0.232). Furthermore, multivariate dispersions did not differ between the groups (week 6, *p* = 0.391; week 12, *p* = 0.377), indicating that the internal variability (dispersion) of the microbial communities was similar between the treatment groups at both time points (Figure 7D).

The microbiological composition of the fecal samples was evaluated at the phylum, family, and genus levels for the 15 most relatively abundant Amplicon Sequence Variants (ASVs) at the end of weeks 6 and 12 of Phase 2 (Figure 8). No difference was observed between the groups at the phylum level in any week. At week 6, at the family level, Erysipelatoclostridiaceae was higher in the CTN group compared to HFD (*p* = 0.0367). Regarding genus, there was a higher abundance of *Massiliomicrobiota* (*p* = 0.0112) and *Lachnospiraceae A2* (*p* = 0.0216) in the CTN compared to HFD. On the other hand, *Berryella* was higher in HFD compared to CTN (*p* = 0.0449). At week 12, at family level, Selenomonadaceae was higher in the HFD group compared to CTN (*p* = 0.0367) and the opposite was found to Eggerthellaceae, which was higher in the CTN group (*p* = 0.0367). In the same week, the genus *Family XIII UCG-001* was higher in the HFD group (*p* = 0.0278). Paired comparisons between week 6 and week 12 (intra-group comparison) revealed no significant difference in the relative abundance at any taxonomic level.

Metabolic and ecological functions were predicted using FAPROTAX (Figure 9A). Twelve functions/categories were identified. A statistically significant difference in relative abundance was observed at Week 12 (*p* = 0.0476), where one category was exclusively found in the HFD group. The bacterium associated with this unique category belongs to the order Rickettsiales, with no family or genus identified. No other intergroup or intragroup differences were found for the remaining functions/categories. Using PICRUSt2, 309 metabolic pathways were identified. Among the 15 most abundant functions (Figure 9B), a statistically significant difference was found for the metabolic pathway PWY-7357 (thiamine diphosphate formation from pyrithiamine and oxythiamine), which was significantly higher in the CTN group compared to the HFD group at week 6 (*p* = 0.0317).

## 4. Discussion

Since we used commercial chow as the base to elaborate the high-fat diet, the nutritional profiles of the diets differed only due to the addition of lard, a source rich in saturated fat widely used in HFD models to promote metabolic changes. This approach provides a model that is easily reproducible in other studies, low-cost, and reduces the variability among the animals [8,28]. The “low-cost” designation to the custom diet refers to the fact that it was prepared from standard chow and lard, with widely accessible ingredients, which typically cost a small fraction of specialized high-fat formulations.

Throughout the induction phase (Phase 1), the HFD group consumed a higher total caloric load than the CTN group due to the caloric density difference between the diets. Regarding the amount ingested, HFD consumption was higher until the 5th week compared to CTN group. After that, there is an equivalence in consumption between the groups, due to the reduction in the intake of the high-fat diet by the animals. This pattern is consistent with previous studies demonstrating that the consumption of rats DIO is like chow-feed and high-fat diets promote weight gain even in the absence of hyperphagia, since fat has a higher energy value [6,10,29,30]. Thus, even small differences in accumulated caloric intake are sufficient to induce obesity in C57BL/6 mice after a few weeks of exposure [4]. In our study, not only did body weight increase with the $45\%$ fat diet, but we also observed increases in abdominal circumference, Lee index, and tail glucose.

Our results contrast with those of de Wit et al., 2008 [31] and Li et al., 2020 [32], who did not find differences in the weight of C57BL/6 mice fed a high-fat diet (53% of fat) compared to controls. However, de Wit et al., 2008 [31] did observe divergent weight gain starting after the second week when using the purified D12451 diet (Research Diets). In another study using 60% of Kcal from fat, mice consumed less mass in grams than the control group, but had a higher caloric intake, resulting in greater weight gain after 12 weeks of HFD [33]. Skalski et al., 2024 [34] compared 45% and 60% fat diets using females C57Bl/6, finding that both groups gained weight, but the time required to achieve a significant difference compared to the controls varied. They noted that only the 60% HFD mice exhibited hyperphagia. Although a 60% fat diet promotes faster weight gain, it is considered an extreme diet that does not represent the human fat intake of the Western diet, which ranges from 36% to 40% fat energy, limiting its relevance for studying human physiology [4].

The elevated Lee index and abdominal circumference observed in our HFD group suggest significant adipose tissue accumulation and body fat redistribution. These markers are widely associated with insulin resistance and increased metabolic risk [35]. Furthermore, the increased blood glucose levels in the HFD animals align with the literature, which links HFD consumption to glucose intolerance and reduced insulin sensitivity, often resulting from adipose tissue inflammation and hepatic oxidative damage as fat storage increases [8,36]. However, even after 12 weeks on a standard diet (Phase 2), the previously obese group maintained a significantly higher Lee index. This finding suggests that partial recovery of obesity is not immediate, and some morphophysiological changes can persist even after returning to a non-obesogenic diet.

During Phase 2, characterized by diet normalization, both groups exhibited weight loss, but the previously obese animals showed greater total weight loss and a larger reduction in abdominal circumference. This observation is consistent with Ji et al., 2023 [37], who reported weight reduction after 12 weeks of standard diet following 21 weeks of DIO, and Yang et al., 2022 [38], who observed favorable effects on weight loss and fat distribution when HFD was withdrawn and replaced with a Chinese-style dietary pattern. These studies collectively demonstrate that HFD withdrawal allows for weight and metabolic recovery.

The significant persistence of higher weights in eWAT, rpWAT and tWAT in the HFD group, even after 12 weeks of standard diet (Phase 2), confirms the effectiveness of the initial obesity model. The accumulation of white adipose tissue (WAT), particularly in the epididymal and retroperitoneal regions, is a classic marker of visceral obesity and insulin resistance in mice [39]. Crucially, the lack of difference in liver, intestine, and colon weights suggests that while the HFD increased fat accumulation, the normalization approach was insufficient to reduce adipose tissue mass to levels comparable to the CTN group, indicating a sustained morphological impact.

At the end of Phase 2, the HFD group displayed lower uric acid, total cholesterol, and non-HDL cholesterol levels compared to the control group. Results regarding cholesterol levels in DIO models are inconsistent [30], as different dietary lipid profiles can modulate hepatic lipid and lipoprotein metabolism differently [40,41]. Furthermore, the animals’ adaptive metabolic state after 12 weeks of dietary normalization may have influenced these values, as lipid metabolism varies according to the stage of obesity development [42]. The reduction in uric acid may be a consequence of weight loss, as some studies identify a reduction in the levels of this metabolite in DIO models after intervention, which also resulted in smaller weight [43,44]. The reduction in total/non-HDL cholesterol and uric acid observed in the HFD group after diet normalization aligns with the metabolic remodeling phase that follows weight loss or caloric restriction. During this period, rapid adjustments in cholesterol transport, lipid turnover, and hepatic metabolism can transiently lower circulating lipid fractions regardless of residual adiposity. Previous studies have shown that caloric restriction in obese conditions improves glucose and lipid [45] and that reductions in circulating cholesterol are driven primarily by weight loss rather than dietary macronutrient composition [46]. Additionally, shifts in gut microbiota and microbial metabolites—particularly increased butyrate—may further modulate lipid handling and systemic inflammation, offering a potential explanation for the improved lipid profile and reduced uric acid levels observed after HFD withdrawal [47]. Uric acid, a sensitive marker of purine metabolism, is known to respond to dietary modulation [48] and tends to decrease in parallel with reductions in hepatic fat content [49]. A limitation of our work is the absence of a biochemical evaluation immediately following the obesity induction period (end of Phase 1). This baseline measurement would have clarified whether the 12 weeks of standard diet (Phase 2) caused a complete reversal of metabolic changes or if the liver was already protected.

We found no evidence of hepatic oxidative stress or changes in antioxidant enzyme activities (CAT and SOD) or oxidative damage markers (TBARS and protein carbonylation) at the end of the study (Phase 2). This suggests that 12 weeks of dietary normalization were sufficient to restore hepatic redox balance, possibly due to the adaptive capacity of hepatic antioxidant defenses and/or the weight loss achieved. The maintenance of stable liver parameters may be related to the liver’s initial adaptive antioxidant capacity, which compensates for increased fatty acid oxidation [50]. Furthermore, weight loss is known to improve liver biochemistry and functionality [51], an effect likely achieved after the lard diet was withdrawn. Tissue antioxidant defenses can be compromised in DIO [52], and the effects of HFD on oxidative stress depend on factors such as exposure duration, fat type, and established obesity status. Dhibi et al. (2011) [53] showed that a HFD containing different trans fatty acids resulted in oxidative stress, and Jarukamjorn et al. (2016) [54] found that mice fed a high-fat/high-fructose (HFHF) diet also increased hepatic TBARS.

The modulation of gut microbiota composition and its metabolic products, such as SCFAs, is a key mechanism linking diet to host metabolic outcomes. We investigated these parameters during the normalization phase, analyzing fecal samples collected at weeks 6 and 12 of standard diet consumption. At Week 6, the CTN group showed higher levels of acetic acid (acetate) compared to the HFD group, while propionic and butyric acids were similar. The lower acetate content in the HFD group suggests that, even after six weeks of dietary normalization, residual metabolic alterations and microbiota remodeling persisted. Acetate is mainly produced by saccharolytic bacteria, particularly Bacteroidetes and Lachnospiraceae members, and serves as a key substrate for lipid and glucose metabolism [55,56]. Its reduction in the HFD group likely reflects a delayed recovery of carbohydrate-fermenting taxa and intestinal fermentation capacity following prolonged exposure to dietary fat. By the end of the normalization phase (Phase 2, week 12), this pattern partially shifted: butyric acid levels (butyrate) were higher in the HFD group, while acetate and propionate became similar between groups. This finding indicates microbial and metabolic adaptation over time, suggesting that certain bacterial groups capable of producing butyrate became more active during recovery. While butyrate is generally beneficial for intestinal health and has demonstrated roles in preventing and treating obesity in animal studies [57], its isolated increase might represent a compensatory fermentative shift rather than a full modulation to eubiosis, particularly when accompanied by altered bacterial composition.

Microbiota results support this interpretation. No differences were detected at the phylum level, suggesting that the overall microbial structure was largely preserved. However, distinct alterations emerged at finer taxonomic resolutions. At week 6, the family Erysipelatoclostridiaceae and the genera *Massiliomicrobiota* and *Lachnospiraceae A2* were more abundant in the CTN group, whereas *Berryella* was enriched in the HFD group. Members of Lachnospiraceae are recognized producers of butyrate and to the formation, one of the possible vias is butyryl-CoA:acetate CoA-transferase, and their persistence in CTN animals aligns with the higher acetate levels observed at this time point [58]. Conversely, the enrichment of *Berryella*, a genus from the family Eggerthellaceae, in the HFD group may be associated with the SCFA production [59].

At the end of Phase 2 (week 12), other differences were observed. The relative abundance of the family Selenomonadaceae was higher in the HFD group compared to CTN and has been previously linked to the development of obesity [60]. In the study of Mo et al., 2022 [61], the relative abundance of this family was reduced in overweight individuals after the supplementation with probiotics. The genus *Family XIII UCG-001* was also higher in the HFD group. This relatively new genus from the Firmicutes phylum had been related to heart failure, being a risk factor and gastric neuroendocrine neoplasms [62]. On the other hand, the Eggerthellaceae family, a polyphenol-degrading and SCFA-producing taxa [63,64], was enriched in the CTN group, suggesting the impact DIO in these bacteria.

The absence of differences in alpha diversity indices (Shannon, Simpson, or richness) between groups or over time suggests that microbial recovery after HFD withdrawal primarily involves compositional and functional remodeling rather than changes in overall species diversity. This agrees with previous findings that the gut microbiota of DIO mice can remain functionally altered for weeks after dietary normalization, even if taxonomic diversity appears restored [65]. Since we used 5 animals per group for metataxonomic evaluation; the low variability of the murine microbiota, as well as the housing in collective cages, the lack of difference between the groups is justified, becoming a limitation in our work.

Regarding intestinal metabolism, the modulation of SCFA over time reflects how dynamic and adaptative gut microbiota could be. The increase in butyrate observed in the HFD group at the last week of the study showed the resilience of microbiota. These metabolic adjustments were accompanied by discrete changes in microbiota composition, notably a higher abundance of *Berryella* in HFD group and enrichment of *Lachnospiraceae A2* and Erysipelatoclostridiaceae in controls. As Lachnospiraceae are recognized butyrate producers, their reduced abundance under DIO conditions may explain the initial drop in butyrate levels, while the later increase suggests a compensatory adaptation involving other bacterial taxa. Considering this study a possible model for others, in a methodological perspective, these findings reinforce that six weeks of intervention are sufficient to detect early microbiota and SCFA shifts associated with metabolic recovery, providing valuable information for refining experimental designs focused on diet normalization and gut metabolic resilience. Although SCFAs play key roles in energy metabolism and intestinal physiology, the temporal shift observed here should be interpreted cautiously, as the study was not designed to determine mechanistic links between SCFAs and the host’s residual metabolic phenotype. Moreover, SCFA concentrations reflect combined effects of microbial composition, substrate availability, colonic absorption, and host metabolism [55,66]. Thus, here SCFAs were evaluated as exploratory outcomes and provide descriptive insights into microbial metabolic activity rather than mechanistic evidence.

The functional prediction analysis using FAPROTAX identified few functions/categories related to the microorganisms found in animal feces. Regarding metabolic functions, we found functions related to the acquisition of energy and carbon by microorganisms, among them chemoheterotrophy; aerobic chemoheterotrophy; fermentation; xylanolysis and ureolysis. Among the ecological categories, we identified human gut; animal parasites or symbionts; intracellular parasites; mammal gut; human pathogens all; human pathogens diarrhea and parasitic symbiotic candidate phyla. An ASV was related to an intracellular parasite present in the HFD group, a member of the order Rickettsiales, an Alphaproteobacterial class of obligate endosymbionts and parasites that comprise mainly intracellular symbiotic eukaryotic bacteria, which can cause diseases in mammals and humans [67,68]. However, due to its low relative abundance and only a single ASV related to this function, the impact would not be significant.

When using the functional prediction analysis using Picrust2, we identify several metabolic pathways. Over the 309 pathways identified, the most abundant were PWY0-1586; PWY-1042; PWY-6317; PWY-8178; P42-PWY; PWY-7791; PWY-7790; PWY-6121; PWY-5686; PWY-7357; PWY-5384; PWY-5097; PWY-6386; PWY-7953; PWY-6387. The identified pathways are mainly grouped into carbohydrate metabolism and precursor production (PWY-1042; PWY-6317; PWY-5384; PWY-8178); energy metabolism (P42-PWY); cofactor and vitamin biosynthesis (PWY-7357; PWY-6121); nucleotide biosynthesis (PWY-5686; PWY-7790; PWY-7791); amino acid biosynthesis (PWY-5097); and bacterial cell wall biosynthesis and maturation (PWY0-1586; PWY-6386; PWY-6387; PWY-7953) according to MetaCyc. The higher abundance of the pathway PWY-7357 (thiamine diphosphate formation from pyrithiamine and oxythiamine) in the CTN group compared with HFD group suggests that DIO alters host–microbe metabolic interactions related to thiamine metabolism. Thiamine diphosphate (TDP) represents the biologically active form of vitamin B1 and serves as an essential cofactor for enzymes involved in oxidative energy metabolism, such as pyruvate dehydrogenase, transketolase, alpha-keto glutaric acid dehydrogenase [69,70]. In metabolically healthy individuals, these pathways are highly active, favoring glucose oxidation and efficient ATP production. Therefore, the greater presence of the PWY-7357 pathway in the CTN group may reflect an enhanced demand for TDP to support oxidative energy metabolism, while evidence shown that obesity may be associated with thiamine deficiency [71]. Although PICRUSt2 identified a different abundance in one pathway between groups, the overall functional predictions should be interpreted cautiously, as the modest taxonomic differences and absence of diversity changes suggest that dietary normalization resulted in only subtle functional modulation rather than broad functional restructuring. It is also important to note that PICRUSt2 and FAPROTAX provide predicted, not directly measured, functional profiles; therefore, their outputs offer insights into potential metabolic trends rather than definitive functional shifts.

Altogether, these findings demonstrate that, following obesity induction, microbiota undergoes subtle functional modulation during the weight loss phase, particularly in the control group, suggesting that dietary normalization can partially restore microbial metabolic balance, although some functions remain influenced by previous dietary patterns.

## 5. Conclusions

Overall, the results confirm that a 12-week high-fat-low-cost diet was effective in inducing increased weight gain and adiposity parameters, reproducing the morphological changes seen in experimental models. The subsequent phase demonstrates the effects of diet normalization and the animals’ metabolic plasticity but also shows that some consequences of prolonged fat consumption persist even after the diet is discontinued. These findings reinforce the applicability of the proposed model for both obesity induction and intervention studies for investigating nutritional, probiotic, or pharmacological interventions aimed at metabolic modulation.

Gut microbiota and metabolite analyses revealed that diet withdrawal led to compositional and functional remodeling rather than complete restoration of the pre-obese state. While alpha diversity remained stable, distinct taxa and SCFA patterns were observed, indicating gradual microbial adaptation. The functional predictions further demonstrated that DIO modulates microbial metabolic capacity, particularly affecting thiamine-related pathways. The higher abundance of the PWY-7357 pathway in control animals suggests a more active thiamine diphosphate synthesis and oxidative carbohydrate metabolism, which were attenuated under HFD conditions. These findings are consistent with evidence that obesity is associated with impaired thiamine utilization and altered microbial contribution to B-vitamin biosynthesis.

Collectively, the results suggest that both the quality and duration of dietary fat exposure drive microbiota functionality and metabolic resilience more than overall diversity. This experimental model of high-fat-low-cost diet provides a robust and reproducible approach for studying obesity development, diet normalization, and interventions targeting host–microbe metabolic interactions.

## Figures and Tables

**Figure 1 nutrients-17-03806-f001:**
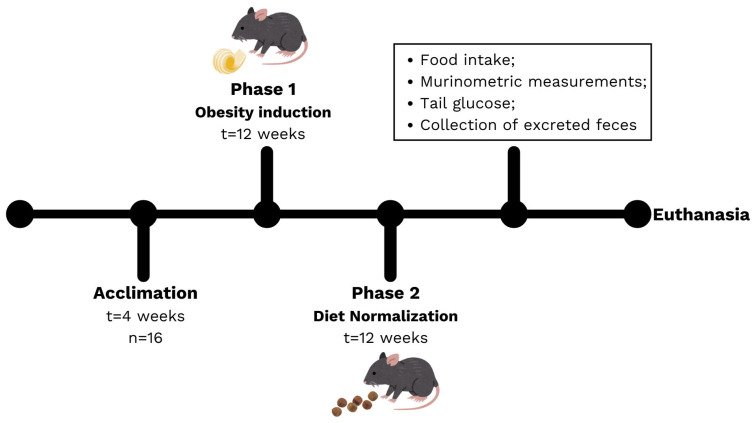
Experimental design, including the acclimation period, Phase 1 (induction) and Phase 2 (diet normalization).

**Figure 2 nutrients-17-03806-f002:**
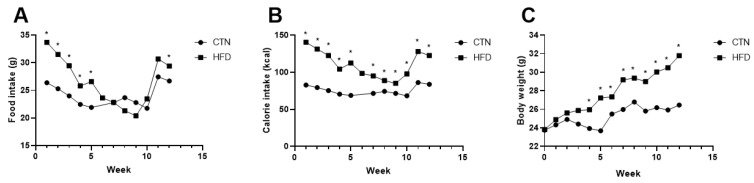
Food; calorie intake and body weight evolution of mice fed with commercial diet (CTN) and mice fed with high-fat, low-cost diet (HFD) for 12 weeks in Phase 1. (**A**) Food intake. (**B**) Calorie intake. (**C**) Evolution of body weight. Values as mean per group. *n* = 8/group. *p* value < 0.05 are shown as * in the figure according to the unpaired *t* test. CTN: control group; HFD: high-fat diet group.

**Figure 3 nutrients-17-03806-f003:**
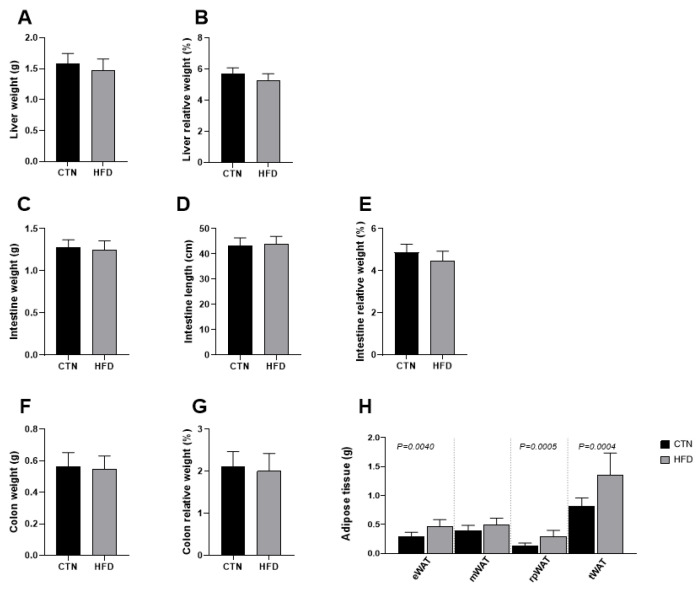
Differences in tissue mass between C57BL-6 mice fed a commercial diet or a low-cost high-fat diet after 12 weeks of diet normalization (**A**) Liver weight; (**B**) Liver relative weight; (**C**) Intestine weight (g); (**D**) Intestine length; (**E**) Intestine relative weight; (**F**) Colon weight; (**G**) Colon relative weight; (**H**) Adipose tissue weight. Values are means ± standard deviation (SD). *n* = 8/group. Values of *p* < 0.05 according to the unpaired *t* test are shown in the figure. CTN: control group; HFD: high-fat diet group; eWAT: epididymal white adipose tissue; mWAT: mesenteric white adipose tissue; rpWAT: retroperitoneal white adipose tissue; tWAT: total white adipose tissue.

**Figure 4 nutrients-17-03806-f004:**
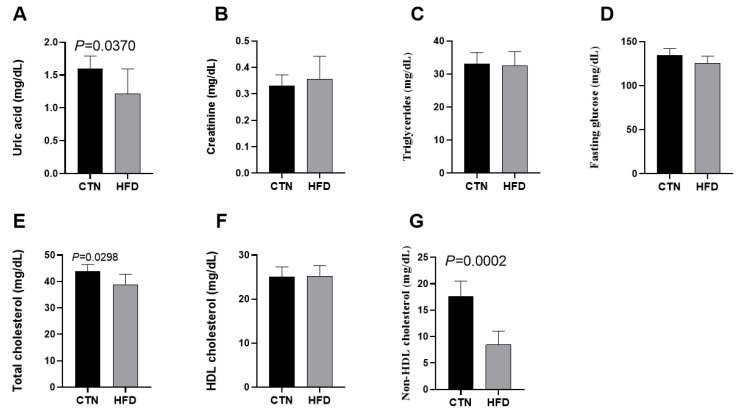
Effects of diet on metabolic variables in C57BL-6 mice after 12 weeks of the obesity induction phase and after 12 weeks of the diet normalization phase. (**A**) Uric acid; (**B**) Creatinine; (**C**) Triglycerides; (**D**) Fasting glucose; (**E**) Total cholesterol; (**F**) HDL cholesterol; (**G**) Non-HDL cholesterol. Values are means ± standard deviation (SD). *n* = 6/group. *p* value < 0.05 are shown in the figure according to the unpaired *t* test. HDL, high-density lipoprotein; CTN: control group; HFD: high-fat diet group.

**Figure 5 nutrients-17-03806-f005:**
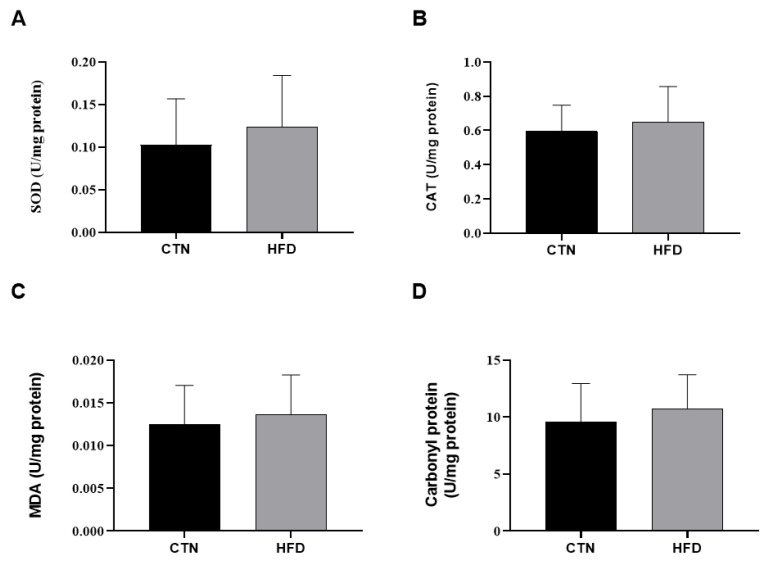
Oxidative stress in liver homogenate of mice after 12 weeks of the obesity induction phase and after 12 weeks of the diet normalization phase. (**A**) Superoxide dismutase; (**B**) Catalase; (**C**) Malondialdehyde; (**D**) Carbonyl protein. Values are means ± standard deviation (SD). *n* = 8/group. *p* value < 0.05 are shown in the figure according to the unpaired *t* test. CTN: control group; HFD: high-fat diet group; SOD: superoxide dismutase; CAT: catalase; MDA: Malondialdehyde.

**Figure 6 nutrients-17-03806-f006:**
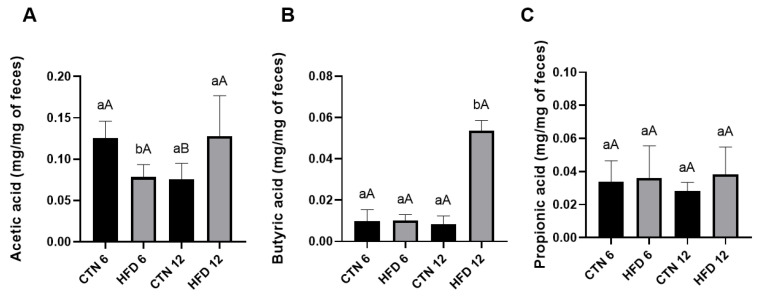
SCFA of excreted feces of mice after 12 weeks of the obesity induction phase and followed by 6 or 12 weeks of the diet normalization phase (**A**) Acetic acid; (**B**) Butyric acid; (**C**) Propionic acid. Values are means ± standard deviation (SD). *n* = 5/group. Small letters refer to statistically significant differences (*p* < 0.05) between groups according to unpaired *t*-test; Capital letters refer to statistically significant differences (*p* < 0.05) between weeks according to paired *t*-test. CTN: control group; HFD: high-fat diet group.

**Figure 7 nutrients-17-03806-f007:**
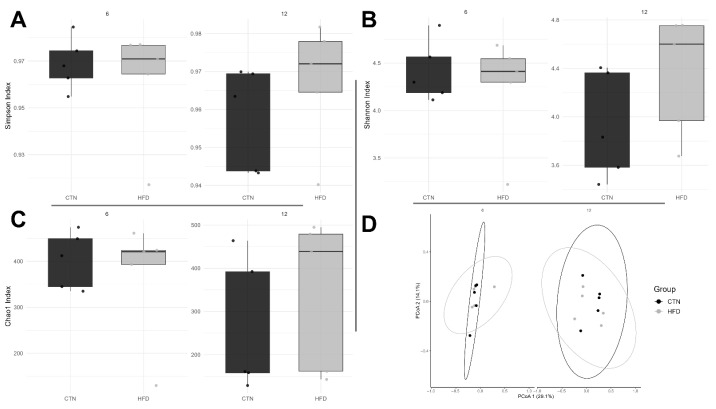
Diversity indices in C57BL-6 mice after 12 weeks of the obesity induction phase and followed by 6 or 12 weeks of the diet normalization phase. Each dot represents a sample. Alpha bacterial diversity: (**A**) Simpson and (**B**) Shannon; (**C**) Bacterial richness; (**D**) Principal Coordinate Analysis (PCoA) plot. *n* = 5/group. CTN: control group; HFD: high-fat diet group.

**Figure 8 nutrients-17-03806-f008:**
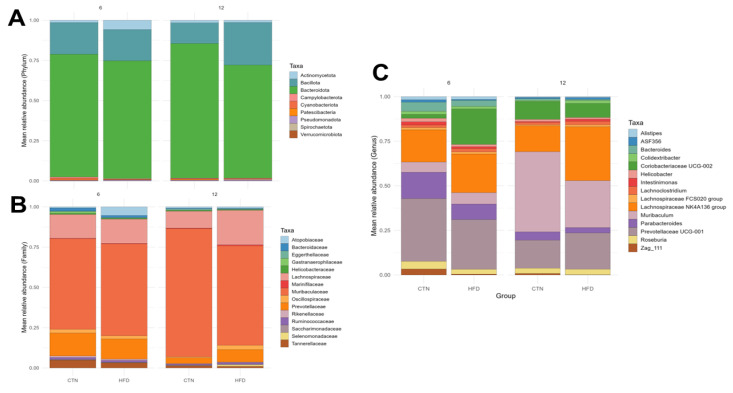
Taxonomy analyses of the feces of mice after 12 weeks of the obesity induction phase and after 6 and 12 weeks of the diet normalization phase. Mean relative abundance. (**A**) Phylum; (**B**) Family; (**C**) Genus. *n* = 5/group. CTN: control group; HFD: high-fat diet group.

**Figure 9 nutrients-17-03806-f009:**
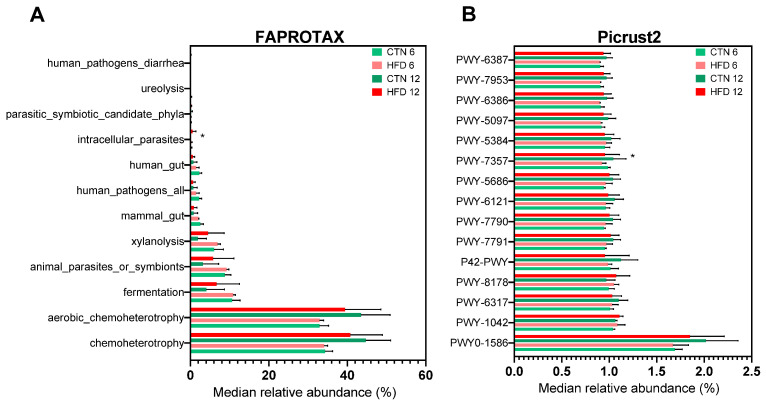
Predicted functions of microbiota by using FAPROTAX and Picrust2 in C57BL-6 mice after 12 weeks of the obesity induction phase and after 6 or 12 weeks of the diet normalization phase. (**A**) FAPROTAX; (**B**) Picrust2. Values are median ± standard deviation (SD). *n* = 5/group. * Refer to statistically significant differences (*p* < 0.05) between groups according to unpaired *t*-test. CTN: control group; HFD: high-fat diet group.

**Table 1 nutrients-17-03806-t001:** Nutritional composition of experimental diets.

Component	AIN93M		HFD	
	g/100 g of Diet	%	g/100 g of Diet	%
Total fat	4.00	11.46	20.90	45.03
Total carbohydrate	45.50	57.96	37.50	35.99
Protein	24.00	30.57	19.80	18.98
Fiber	5.00	-	5.00	-
Ashes	9.00	-	9.00	-
Calories (kcal/100 g of diet)	314	-	417	-

**Table 2 nutrients-17-03806-t002:** Murinometric measurements of the animals after 12 weeks of the obesity induction phase and after 12 weeks of the diet normalization phase.

Parameter	Phase 1			Phase 2		
	CTN	HFD	*p* Value	CTN	HFD	*p* Value
Final body weight [g]	28.12 ± 1.40	30.42 ± 1.97	**0.0416**	27.12 ± 1.70	27.76 ± 1.14	0.4197
Δ Weight [g]	2.08 ± 1.35	7.18 ± 2.43	**0.0004**	−0.93 ± 2.27	−3.84 ± 2.28	**0.0338**
Final abdominal circumference [cm]	7.23 ± 0.34	7.63 ± 0.30	**0.0375**	6.51 ± 0.44	6.46 ± 0.24	0.7837
Δ abdominal circumference [cm]	-	-	**-**	−0.38 ± 0.38	−1.03 ± 0.24	**0.0033**
Lee index	3.37 ± 0.16	3.89 ± 0.13	**0.0126**	3.67 ± 0.07	3.92 ± 0.10	**<0.0001**
Tail blood glucose [mg/dL]	120.0 ± 11.52	142.9 ± 9.83	**0.0011**	149.3 ± 9.55	149.2 ± 11.99	0.9844

Values are means ± standard deviation (SD). *n* = 8/group. Δ Weight (Final weight–Initial weight). Δ abdominal circumference (Final abdominal circumference- Initial abdominal perimeter). Values of *p* < 0.05 according to the unpaired *t* test are marked in bold. CTN: control group; HFD: high-fat diet group.

## Data Availability

The original contributions presented in this study are included in the article. Further inquiries can be directed to the corresponding author.

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
