# Peer review of "A Low-Cost, High-Fat Diet Effectively Induces Obesity and Metabolic Alterations and Diet Normalization Modulates Microbiota in C57BL/6 Mice"

_nutrients, 2025, doi:10.3390/nu17233806_

Round 1
Reviewer 1 Report
Comments and Suggestions for Authors
This manuscript presents a well-designed experimental study evaluating (1) the efficiency of a low-cost, lard-based high-fat diet (HFD) to induce obesity in C57BL/6 mice, and (2) the extent to which metabolic, oxidative, and microbiota-related alterations can be reversed by returning the animals to a standard diet. The study spans 24 weeks in total and includes extensive metabolic, biochemical, SCFA, oxidative stress, and 16S rRNA microbiota analyses. Overall, the study is promising but requires minor revisions before being suitable for publication, including:
-
At the end of Phase 2, HFD animals displayed lower uric acid and total/non-HDL cholesterol than controls, despite having more adipose tissue. This paradox requires more discussion: Is this an adaptive phase of metabolic remodeling after weight loss? Could caloric restriction disproportionately affect lipid metabolism in previously obese animals? Are these findings consistent with known effects of lard-based HFD withdrawal?
-
The SCFA section shows a temporal shift:
-
Week 6: CTN → higher acetate
-
Week 12: HFD → higher butyrate
However, the microbiota taxa do not clearly map onto these shifts. For example, at Week 12, butyrate-producing Lachnospiraceae are not higher in HFD, yet butyrate is. Clarify possible mechanistic explanations or acknowledge inconsistency between taxa and SCFA outputs.
-
-
While common in exploratory microbiota studies, the small sample size drastically reduces statistical power for:
-
Alpha and beta diversity
-
Differential taxa abundance
-
Functional predictions (FAPROTAX and PICRUSt2)
The manuscript should explicitly state how this limitation affects interpretation, especially given the subtle differences reported at genus/family levels.
-
Author Response
This manuscript presents a well-designed experimental study evaluating (1) the efficiency of a low-cost, lard-based high-fat diet (HFD) to induce obesity in C57BL/6 mice, and (2) the extent to which metabolic, oxidative, and microbiota-related alterations can be reversed by returning the animals to a standard diet. The study spans 24 weeks in total and includes extensive metabolic, biochemical, SCFA, oxidative stress, and 16S rRNA microbiota analyses. Overall, the study is promising but requires minor revisions before being suitable for publication, including:
1- At the end of Phase 2, HFD animals displayed lower uric acid and total/non-HDL cholesterol than controls, despite having more adipose tissue. This paradox requires more discussion: Is this an adaptive phase of metabolic remodeling after weight loss? Could caloric restriction disproportionately affect lipid metabolism in previously obese animals? Are these findings consistent with known effects of lard-based HFD withdrawal?
We thank the reviewer for the comment, and we agree that this requires further discussion. The post–weight loss or caloric restriction phase is characterized by a period of metabolic remodeling in which cholesterol transport, lipid turnover, and hepatic metabolism undergo rapid adjustments. These adaptations can transiently reduce circulating lipid fractions independently of the remaining adipose mass. Kirchner et al. (2012) demonstrate that caloric restriction in the context of obesity improves both glucose and lipid homeostasis, and Minderis et al. (2021) report that reductions in circulating cholesterol are driven primarily by weight loss, rather than by macronutrient composition. This supports the idea that caloric restriction may disproportionately enhance lipid metabolism in previously obese animals.
Moreover, alterations in gut microbiota and microbial metabolite production (particularly butyrate) may contribute to systemic metabolic shifts. These changes could partly explain the reduced plasma lipid levels and lower uric acid in the HFD group after diet normalization, as microbial-derived metabolites are known to modulate lipid handling and inflammation (Gong et al., 2025). Uric acid itself, a marker of purine catabolism, is sensitive to dietary patterns (Liu et al., 2015) and has been shown to decrease in parallel with reductions in hepatic fat content (Mishima et al., 2025).
We expanded this discussion in the manuscript, please see page 14, lines 437-449.
Kirchner, H., Hofmann, S. M., Fischer-Rosinský, A., Hembree, J., Abplanalp, W., Ottaway, N., ... & Habegger, K. M. (2012). Caloric restriction chronically impairs metabolic programming in mice. Diabetes, 61(11), 2734-2742.
Minderis, P., Fokin, A., Dirmontas, M., Kvedaras, M., & Ratkevicius, A. (2021). Caloric restriction per se rather than dietary macronutrient distribution plays a primary role in metabolic health and body composition improvements in obese mice. Nutrients, 13(9), 3004.
Gong, H., Zuo, H., Wu, K., Gao, X., Lan, Y., & Zhao, L. (2025). Systemic and Retinal Protective Effects of Butyrate in Early Type 2 Diabetes via Gut Microbiota–Lipid Metabolism Interaction. Nutrients, 17(14), 2363.
Liu, Z., Que, S., Zhou, L., & Zheng, S. (2015). Dose-response relationship of serum uric acid with metabolic syndrome and non-alcoholic fatty liver disease incidence: a meta-analysis of prospective studies. Scientific Reports, 5(1), 14325.
Mishima, M. D. V., Bernardes, A. L., Pinheiro, C. A., Lana, L. G., Campos, I. X., de Lana, V. S., ... & do Carmo Gouveia Peluzio, M. (2025). Cashew Nut Oil Improves Lipid Metabolism and Fat Liver Deposition in High‐Fat Diet‐Fed C57BL/6J Mice. Lipids.
2- The SCFA section shows a temporal shift:
- Week 6: CTN → higher acetate
- Week 12: HFD → higher butyrate
However, the microbiota taxa do not clearly map onto these shifts. For example, at Week 12, butyrate-producing Lachnospiraceae are not higher in HFD, yet butyrate is. Clarify possible mechanistic explanations or acknowledge inconsistency between taxa and SCFA outputs.
We thank the reviewer for noting the apparent inconsistency between taxonomic shifts and fecal SCFA profiles. The discrepancy between SCFA levels and the abundance of producing microbial families can occur through various mechanisms that do not depend exclusively on the relative abundance of taxa. Different bacterial groups can produce the same SCFAs even when classical producers (e.g., Lachnospiraceae) are not increased. Butyrate is an important metabolite produced through gastrointestinal microbial fermentation and is used as an energy source to the colonic mucosa (Fu et al., 2019) and it can rise despite stable or lower abundance of typical butyrate-producing taxa. In addition to production by fermentation of dietary fibers, primarily by bacteria of the Clostridium cluster of the phylum Firmicutes, butyrate can also be produced through the conversion of lactate and acetate to butyrate via the acetyl-CoA pathway (Güler et al., 2025; Singh et al., 2023).
Even with a similar microbiota, altering the diet composition, with greater availability of fiber and other substrates from the commercial diet after removing the lard that was included in the HFD diet formulation, can alter metabolic fluxes, increasing SCFAs without detectable taxonomic changes (Den Besten et al., 2013). The gut microbiota can also adapt to nutritional changes, such as the reduction of fat amount that happened in Phase 2, in order to maintain the production of essential metabolites such as SCFAs (Koh et al., 2016).
We detail this discussion in the manuscript, please see page 16, lines 528-534.
Fu, X., Liu, Z., Zhu, C., Mou, H., & Kong, Q. (2019). Nondigestible carbohydrates, butyrate, and butyrate-producing bacteria. Critical reviews in food science and nutrition, 59(sup1), S130-S152.
Singh, V., Lee, G., Son, H., Koh, H., Kim, E. S., Unno, T., & Shin, J. H. (2023). Butyrate producers,“The Sentinel of Gut”: Their intestinal significance with and beyond butyrate, and prospective use as microbial therapeutics. Frontiers in microbiology, 13, 1103836.
Güler, M. S., Arslan, S., Ağagündüz, D., Cerqua, I., Pagano, E., Canani, R. B., & Capasso, R. (2025). Butyrate: A potential mediator of obesity and microbiome via different mechanisms of actions. Food Research International, 199, 115420.
Den Besten, G., Van Eunen, K., Groen, A. K., Venema, K., Reijngoud, D. J., & Bakker, B. M. (2013). The role of short-chain fatty acids in the interplay between diet, gut microbiota, and host energy metabolism. Journal of lipid research, 54(9), 2325-2340.
Koh, A., De Vadder, F., Kovatcheva-Datchary, P., & Bäckhed, F. (2016). From dietary fiber to host physiology: short-chain fatty acids as key bacterial metabolites. Cell, 165(6), 1332-1345.
3- While common in exploratory microbiota studies, the small sample size drastically reduces statistical power for:
- Alpha and beta diversity
- Differential taxa abundance
- Functional predictions (FAPROTAX and PICRUSt2)
The manuscript should explicitly state how this limitation affects interpretation, especially given the subtle differences reported at genus/family levels.
We thank the reviewer for this observation. In this study, microbiota analyses were performed using five samples per group in two different times, a number selected due to the substantial cost associated with sequencing and the lower variability of lab rodent’s microbiota in comparison with humans. We acknowledge that this sample size is smaller than ideal for capturing subtle shifts in microbial community structure and may reduce statistical power for alpha and beta diversity metrics, differential taxonomic abundance, and functional predictions.
According to Moore and Stanley (2016), for a more in-depth study of microbiota with high power, an n>50 animals would be necessary, which is unfeasible considering an exploratory study and respecting the 3Rs principle. However, similar sample sizes (n = 5-7 per group) are common and accepted, where controlled environmental and genetic conditions contribute to markedly lower inter-individual variability compared to human cohorts (Ricci et al., 2020; Richter et al., 2018). Many studies analyzing 5-6 C57BL6/J animals per group have been published (Mamun et al., 2025; Baek et al., 2025; Zhu; Hou, 2024; Wang et al., 2023).
To minimize bias arising from limited sample size, we applied stringent quality-filtering criteria (prevalence and abundance thresholds), corrected statistical analyses using FDR when appropriate, and restricted pathway-level functional comparisons to the most abundant features to avoid inflation of false positives. Even so, the results involving microbial taxa or pathways with small effect sizes should be interpreted cautiously and considered exploratory. Future studies with larger cohorts are warranted to validate and expand upon these findings.
We added this point as a limitation in our work (please see page 16, lines 512-515).
Richter, V., Muche, R., & Mayer, B. (2018). How much confidence do we need in animal experiments? Statistical assumptions in sample size estimation. Journal of Applied Animal Welfare Science, 21(4), 325-333.
Ricci, C., Baumgartner, J., Malan, L., & Smuts, C. M. (2020). Determining sample size adequacy for animal model studies in nutrition research: limits and ethical challenges of ordinary power calculation procedures. International journal of food sciences and nutrition, 71(2), 256-264.
Moore, R. J., & Stanley, D. (2016). Experimental design considerations in microbiota/inflammation studies. Clinical & Translational Immunology, 5(7), e92.
Zhu, H., & Hou, T. (2024). Modulatory Effects of lactarius hatsudake on obesity and gut microbiota in high-fat diet-fed C57BL/6 mice. Foods, 13(6), 948.
Wang, Z., Liu, H., Song, G., Gao, J., Xia, X., & Qin, N. (2023). Cherry juice alleviates high-fat diet-induced obesity in C57BL/6J mice by resolving gut microbiota dysbiosis and regulating microRNA. Food & Function, 14(6), 2768-2780.
Mamun, M. A. A., Rakib, A., Mandal, M., & Singh, U. P. (2025). Impact of a high-fat diet on the gut microbiome: a comprehensive study of microbial and metabolite shifts during obesity. Cells, 14(6), 463.
Baek, K. W., Yun, K., He, M. T., Kim, N. K., Kim, J. S., Ahn, K., & Kim, J. H. (2025). Integrated transcriptomic and 16S rRNA analyses reveal colon and brain barrier-preserving effects of red radish (Raphanus sativus L.) sprout supplementation in high-fat diet-fed mice. Genes & Genomics, 1-15.

Reviewer 2 Report
Comments and Suggestions for Authors
Reviewer comments
- The absence of metabolic, oxidative stress, and microbiota data at the end of Phase 1 (obesity induction) severely limits the interpretation of the "reversal" or "restoration" claimed in Phase 2.
- The sample size (n=8) was justified only for weight gain. The study is likely underpowered to detect significant differences in the primary endpoints of microbiota composition, SCFAs, and oxidative markers.
- The "caloric restriction" phase is a misnomer. The protocol describes an ad libitum dietary switch, not controlled food intake, confounding the interpretation of the results.
- The conclusions about "functional restructuring" are overstated given the minimal significant taxonomic differences, absence of diversity changes, and only one significant metabolic pathway.
- The biological relevance of the temporal SCFA shift (acetate to butyrate) is speculative and not mechanistically linked to the host's persistent metabolic phenotype.
- The decision to forego multiple testing correction for the PICRUSt2 analysis (top 15 pathways) is statistically unsound and increases the risk of false-positive findings.
- The "low-cost" advantage is not quantitatively demonstrated through a cost comparison with commercial high-fat diets.
- The study is descriptive and correlative, failing to provide mechanistic links between the specific microbial changes and the sustained metabolic alterations (e.g., higher Lee index, adipose mass).
- Key details for microbiota analysis (e.g., sequencing platform, final sequence depth/reads per sample, specific software versions) are missing, hindering reproducibility.
- The finding of improved lipid profiles (lower cholesterol) in the previously obese group is counterintuitive and not sufficiently explained, requiring deeper investigation.
Comments on the Quality of English Language
The manuscript requires moderate English editing by a native speaker or professional language service to correct the accumulated minor errors and refine the phrasing, which will significantly enhance its clarity and professionalism.
Author Response
1. The absence of metabolic, oxidative stress, and microbiota data at the end of Phase 1 (obesity induction) severely limits the interpretation of the "reversal" or "restoration" claimed in Phase 2.
We thank the reviewer for this observation. We agree that collecting metabolic, oxidative stress, and microbiota data at the end of Phase 1 would have strengthened the longitudinal interpretation of the reversal process. However, collecting intermediate metabolic data would have required additional blood sampling, which is a stressful procedure for mice, and oxidative stress analyses depended on hepatic tissue, which can only be obtained after euthanasia. To avoid additional distress and to minimize the number of animals used, all metabolic, oxidative stress, and microbiota assessments were therefore performed only after the completion of Phase 2.
Our primary aim in this study was to evaluate whether a low-cost HFD with 45% of fat could induce obesity and despite the lack of intermediate sampling, the effects of the HFD during Phase 1 are supported by several robust indicators measured at baseline (week 0) and at the transition between phases, including body weight, adiposity, Lee index, abdominal circumference, and fasting glycemia—all of which demonstrated clear obesity induction compared to controls. These outcomes are consistent with the well-established metabolic alterations induced by lard-based HFDs in C57BL/6 mice, as described in the literature.
We’ve presented this limitation in our work (please see page 14, lines 449-453) and also, to clarify this point, we have revised the text, and we replaced the use of "reversal" and “restoration" to “modulation”; “recovery” and “normalization”, highlighted in red in the text.
He, M. Q., Wang, J. Y., Wang, Y., Sui, J., Zhang, M., Ding, X., ... & Shi, B. Y. (2020). High-fat diet-induced adipose tissue expansion occurs prior to insulin resistance in C57BL/6J mice. Chronic Diseases and Translational Medicine, 6(3), 198-207.
Ji, T., Fang, B., Wu, F., Liu, Y., Cheng, L., Li, Y., ... & Zhu, L. (2023). Diet change improves obesity and lipid deposition in high-fat diet-induced mice. Nutrients, 15(23), 4978.
2. The sample size (n=8) was justified only for weight gain. The study is likely underpowered to detect significant differences in the primary endpoints of microbiota composition, SCFAs, and oxidative markers.
We appreciate this observation. Our sample size calculation was indeed based on body weight gain, as this was the primary phenotypic indicator for validating the HFD-induced obesity. As we answered to the reviewer 1, similar HFD-induced obesity and microbiota studies commonly use group sizes ranging from 5 to 10 animals, particularly when sequencing costs and ethical considerations limit sample expansion. Our sample size falls within the typical range reported for exploratory microbiota analyses in murine models, and microbiota analyses employ techniques—such as PERMANOVA, ordination, and relative abundance comparisons—that are widely used in studies with similar sample sizes and remain valid for detecting biologically meaningful differences.
We added this point as a limitation in our work (please see page 16, lines 512-515).
Mamun, M. A. A., Rakib, A., Mandal, M., & Singh, U. P. (2025). Impact of a high-fat diet on the gut microbiome: a comprehensive study of microbial and metabolite shifts during obesity. Cells, 14(6), 463.
Xue, Y., Tang, R., & Liu, Z. (2025). Inulin Alleviates Intestinal Barrier Dysfunction Induced by Chronic Intermittent Hypoxia by modulating intestinal microbiota in Mice. American Journal of Physiology-Regulatory, Integrative and Comparative Physiology.
Baek, K. W., Yun, K., He, M. T., Kim, N. K., Kim, J. S., Ahn, K., & Kim, J. H. (2025). Integrated transcriptomic and 16S rRNA analyses reveal colon and brain barrier-preserving effects of red radish (Raphanus sativus L.) sprout supplementation in high-fat diet-fed mice. Genes & Genomics, 1-15.
Mishima, M. D. V., Da Silva, B. P., Gomes, M. J. C., Toledo, R. C. L., Mantovani, H. C., José, V. P. B. D. S., ... & Martino, H. S. D. (2022). Effect of chia (Salvia hispanica L.) associated with high-fat diet on the intestinal health of wistar rats. Nutrients, 14(22), 4924.
3. The "caloric restriction" phase is a misnomer. The protocol describes an ad libitum dietary switch, not controlled food intake, confounding the interpretation of the results.
We appreciate your comment and agree that it may cause confusion in the interpretation of the results. To avoid confusion, we have changed the term "calorie restriction" to "diet normalization", highlighted in red in the text. We’ve also changed in Figure 1 (please see page 4, lines 108)
4. The conclusions about "functional restructuring" are overstated given the minimal significant taxonomic differences, absence of diversity changes, and only one significant metabolic pathway.
We appreciate this comment and agree that the term “functional restructuring” may imply a broader or more robust shift than what our data support. Our analyses detected only a few taxonomic differences, no significant changes in alpha or beta diversity, and a single metabolic pathway differing between groups; therefore, the conclusion should indeed be more conservative. We have revised the manuscript to temper the interpretation of the microbiota results. We replaced “functional restructuring” with a more appropriate term: “subtle functional modulation”, please see page 17, line 574.
Additionally, we emphasized in the Discussion that the modest functional signal aligns with the minimal taxonomic differences observed, and that the absence of diversity changes indicates that diet normalization produced modest microbiota alterations, rather than a broad restructuring (please see page 17, lines 566-572).
5. The biological relevance of the temporal SCFA shift (acetate to butyrate) is speculative and not mechanistically linked to the host's persistent metabolic phenotype.
We thank the reviewer for this comment. We agree that the biological interpretation of the temporal shift in SCFAs (higher acetate in CTN and higher butyrate in HFD mice at the end of Phase 2) should be presented with caution. Our study design does not allow for a mechanistic link between these SCFA profiles, and the persistent metabolic differences observed after diet normalization. Therefore, we have tempered our interpretation by describing these results as associative observations, rather than mechanistic insights. Accordingly, we have updated the Discussion in the revised manuscript. We clarify that SCFAs were evaluated as exploratory endpoints and that the observed acetate–butyrate shift may reflect diet-driven microbial adjustments rather than direct drivers of the host metabolic phenotype (please see page 16, lines 528-534).
6. The decision to forego multiple testing correction for the PICRUSt2 analysis (top 15 pathways) is statistically unsound and increases the risk of false-positive findings.
Thank you for this important statistical observation. The multiple-testing correction is essential in high-dimensional functional prediction analyses. In the revised version, we clarified two key points.
Although our initial justification referred to visualisation of the top 15 most abundant pathways, the inferential statistics in our PICRUSt2 workflow already included FDR correction (Benjamini–Hochberg) for all comparisons. The selection of the top 15 most abundant pathways was only used for figure readability.
The second one is that after applying outlier removal (IQR method) and recomputing all statistics with FDR correction, only one of the pathways, KOs or ECs was significantly different between groups. This reinforces the robustness of our findings and reduces the likelihood of type I errors.
We have made changes in the methodology to clarify it (please see page 6, lines 218-226).
7. The "low-cost" advantage is not quantitatively demonstrated through a cost comparison with commercial high-fat diets.
We thank the reviewer for this comment. Our intention in using the term “low-cost” was to emphasize that the diet was formulated using only standard chow and lard, which are considerably less expensive than purified commercial HFDs and their corresponding shift costs. Because prices vary substantially across regions and suppliers, a precise quantitative comparison may not be directly generalizable and was therefore not included. To address the reviewer’s concern, we have now clarified in the manuscript that the “low-cost” designation refers to the use of readily available ingredients, which typically cost a small fraction of specialized high-fat formulations. We have revised the text to highlight the relative affordability and accessibility, rather than presenting an exact price comparison (please see page 13, lines 376-378).
8. The study is descriptive and correlative, failing to provide mechanistic links between the specific microbial changes and the sustained metabolic alterations (e.g., higher Lee index, adipose mass).
We thank the reviewer for this point. We agree that our study is descriptive and does not allow mechanistic inferences regarding causal relationships between the microbial differences and the persistent metabolic alterations observed after diet normalization. Our analyses were designed to characterize metabolic and microbiota changes associated with dietary intervention, and not to experimentally determine causation.
9. Key details for microbiota analysis (e.g., sequencing platform, final sequence depth/reads per sample, specific software versions) are missing, hindering reproducibility.
We thank the reviewer for this observation. We’ve changed the methodology section (please see page 6, lines 185-226) and results (please see page 11, lines 321-322) to clarify these points and allow reproducibility
10. The finding of improved lipid profiles (lower cholesterol) in the previously obese group is counterintuitive and not sufficiently explained, requiring deeper investigation.
We thank the reviewer for the comment, and we agree that the finding is counterintuitive and requires further discussion. As we answered to the reviewer 1, the post–weight loss or caloric restriction phase is characterized by a period of metabolic remodeling in which cholesterol transport, lipid turnover, and hepatic metabolism undergo rapid adjustments. These adaptations can transiently reduce circulating lipid fractions independently of the remaining adipose mass. Kirchner et al. (2012) demonstrate that caloric restriction in the context of obesity improves both glucose and lipid homeostasis, and Minderis et al. (2021) report that reductions in circulating cholesterol are driven primarily by weight loss, rather than by macronutrient composition. This supports the idea that caloric restriction may disproportionately enhance lipid metabolism in previously obese animals.
Moreover, alterations in gut microbiota and microbial metabolite production (particularly butyrate) may contribute to systemic metabolic shifts. These changes could partly explain the reduced plasma lipid levels and lower uric acid in the HFD group after diet normalization, as microbial-derived metabolites are known to modulate lipid handling and inflammation (Gong et al., 2025). Uric acid itself, a marker of purine catabolism, is sensitive to dietary patterns (Liu et al., 2015) and has been shown to decrease in parallel with reductions in hepatic fat content (Mishima et al., 2025).
We expanded this discussion in the manuscript, please see page 14, lines 437-449.
Kirchner, H., Hofmann, S. M., Fischer-Rosinský, A., Hembree, J., Abplanalp, W., Ottaway, N., ... & Habegger, K. M. (2012). Caloric restriction chronically impairs metabolic programming in mice. Diabetes, 61(11), 2734-2742.
Minderis, P., Fokin, A., Dirmontas, M., Kvedaras, M., & Ratkevicius, A. (2021). Caloric restriction per se rather than dietary macronutrient distribution plays a primary role in metabolic health and body composition improvements in obese mice. Nutrients, 13(9), 3004.
Gong, H., Zuo, H., Wu, K., Gao, X., Lan, Y., & Zhao, L. (2025). Systemic and Retinal Protective Effects of Butyrate in Early Type 2 Diabetes via Gut Microbiota–Lipid Metabolism Interaction. Nutrients, 17(14), 2363.
Liu, Z., Que, S., Zhou, L., & Zheng, S. (2015). Dose-response relationship of serum uric acid with metabolic syndrome and non-alcoholic fatty liver disease incidence: a meta-analysis of prospective studies. Scientific Reports, 5(1), 14325.
Mishima, M. D. V., Bernardes, A. L., Pinheiro, C. A., Lana, L. G., Campos, I. X., de Lana, V. S., ... & do Carmo Gouveia Peluzio, M. (2025). Cashew Nut Oil Improves Lipid Metabolism and Fat Liver Deposition in High‐Fat Diet‐Fed C57BL/6J Mice. Lipids.
Comments on the Quality of English Language
The manuscript requires moderate English editing by a native speaker or professional language service to correct the accumulated minor errors and refine the phrasing, which will significantly enhance its clarity and professionalism.
We appreciate and thank the reviewer’s observation regarding the quality of the English language. We fully agree that professional editing will improve clarity and readability. To ensure an efficient and consistent revision, we plan to perform the full language polishing after implementing all reviewer comments and editorial adjustments, so that the manuscript undergoes a single, comprehensive round of English editing.

Round 2
Reviewer 2 Report
Comments and Suggestions for Authors
Accept in present form